

# *RIdeogram*: drawing SVG graphics to visualize and map genome-wide data on the idiograms

Zhaodong Hao[1,2], Dekang Lv[3], Ying Ge[3], Jisen Shi[1], Dolf Weijers[2], Guangchuang Yu[4] and Jinhui Chen[1]

[1] Key Laboratory of Forest Genetics & Biotechnology of Ministry of Education, Co-Innovation Center for Sustainable Forestry in Southern China, Nanjing Forestry University, Nanjing, Jiangsu, China
[2] Laboratory of Biochemistry, Wageningen University, Wageningen, Haarlem, Netherlands
[3] Institute of Cancer Stem Cell, Dalian Medical University, Dalian, Liaoning, China
[4] Institute of Bioinformatics, School of Basic Medical Sciences, Southern Medical University, Guangzhou, Guangdong, China

## ABSTRACT

**Background**. Owing to the rapid advances in DNA sequencing technologies, whole genome from more and more species are becoming available at increasing pace. For whole-genome analysis, idiograms provide a very popular, intuitive and effective way to map and visualize the genome-wide information, such as GC content, gene and repeat density, DNA methylation distribution, genomic synteny, etc. However, most available software programs and web servers are available only for a few model species, such as human, mouse and fly, or have limited application scenarios. As more and more non-model species are sequenced with chromosome-level assembly being available, tools that can generate idiograms for a broad range of species and be capable of visualizing more data types are needed to help better understanding fundamental genome characteristics.

**Results**. The R package *RIdeogram* allows users to build high-quality idiograms of any species of interest. It can map continuous and discrete genome-wide data on the idiograms and visualize them in a heat map and track labels, respectively.

**Conclusion**. The visualization of genome-wide data mapping and comparison allow users to quickly establish a clear impression of the chromosomal distribution pattern, thus making *RIdeogram* a useful tool for any researchers working with omics.

Corresponding authors
Guangchuang Yu, gcyu1@smu.edu.cn
Jinhui Chen, chenjh@njfu.edu.cn

## INTRODUCTION

Recently, with the development of sequencing technologies, especially rapid advances in third generation sequencing including Pacific Biosciences (*Eid et al., 2009*) and Oxford Nanopore Technologies (*Laver et al., 2015*), BioNano genome mapping (*Cao et al., 2014*) and high-throughput chromatin conformation capture sequencing (*Dekker et al., 2002*), more and more species have their genomes sequenced or updated to the chromosome level (*Jiao & Schneeberger, 2017*; *Phillippy, 2017*). After the chromosome-level genome completion, an overview of some genome characteristics can help to better understand a

species genome, such as gene and transposon distribution across the sunflower genome (*Badouin et al., 2017*).

An idiogram, also known as a karyotype, is defined as the phenotypic appearance of chromosomes in the nucleus of an eukaryotic cell and has been widely used to visualize the genome-wide data since the first web server, *Idiographica*, came online in 2007 (*Kin & Ono, 2007*). There are dozens of tools have been developed for circular genome visualization with a Perl language-based tool *Circos* being the most used one (*Krzywinski et al., 2009*; *Parveen, Khurana & Kumar, 2019*). In contrast, there are not many alternatives for non-circular plots of whole genome information on idiograms. Although few R packages, like *GenomeGraphs* (*Durinck et al., 2009*), *ggbio* (*Yin, Cook & Lawrence, 2012*), *IdeoViz* (*Pai & Ren, 2014*), *chromPlot* (*Orostica & Verdugo, 2016*) and *chromDraw* (*Janecka & Lysak, 2016*), and JavaScript libraries, like *Ideogram.js* (*Weitz et al., 2017*) and *karyotypeSVG* (*Prlic, 2017*), have been developed for non-circular genome visualization, they are either limited in several species and data visualization types or lacking the ample customization. Recently, two R packages, *karyoploteR* (*Gel & Serra, 2017*) and *chromoMap* (*Anand, 2019*), with strengthened capacities have been developed.

However, one function that all these non-circular plots fail to achieve, as *Circos* does, is to visualize the relationship between two or more species using Bezier curves on idiograms. This function is very useful and allows to interpret genome-wide relationships more intuitively, especially in the visualization of whole genome duplication. Indeed, *Circos* is usually used to show syntenic blocks both in inter- and intraspecies genome comparisons using Bezier curves (*Hu et al., 2019*; *Wang et al., 2019*). Thus, there is a lack of a R package for non-circular genome visualization and allowing to visualize genome-wide relationships between two or more species using Bezier curves on idiograms.

Scalable Vector Graphics (SVG) is a language for describing two-dimensional graphics applications and images. SVG graphics is defined in an eXtensible Markup Language (XML) text file which means that one can easily use any text editor or drawing software to create and edit SVG graphics. Most R graphics packages are built on two graphics systems, the traditional graphics system and the grid graphics system. Here, we developed an R package (*RIdeogram*) to draw high-quality idiograms without species limitations, that allows to visualize and map whole-genome information on the idiograms based on the SVG language. Besides, *RIdeogram* can also be used to show the genome synteny with Bezier curves linking the syntenic blocks on idiograms.

## DESCRIPTION

The package *RIdeogram* is written in R (*R Core Team, 2018*), one of the most popular programming languages widely used in statistical computing, data analytics and graphics. However, this new R graphics package is not built based on any existing graphics systems. We use the R environment to read the custom input files and calculate the drawing element positions in a coordinate system. Then, we use R to write all element information into a text file following the XML format which are used to define graphics by the SVG language. A list of the currently implemented commands is given in Table 1. In general, there are three

**Table 1  Functions contained in the package *RIdeogram*.**

| Function name | Description |
| --- | --- |
| GFFex | Extract information from a GFF3 format genome annotation fill |
| ideogram | Map and visualize the genome-wide data on the idiograms |
| convertSVG | Convert the output file from the SVG format to the format users chose |
| svg2tiff | Convert the output file from the SVG format to the TIFF format |
| svg2pdf | Convert the output file from the SVG format to the PDF format |
| svg2jpg | Convert the output file from the SVG format to the JPG format |
| svg2png | Convert the output file from the SVG format to the PNG format |

main functions, *GFFex*, *ideogram* and *convertSVG* implemented in the package *RIdeogram*. Users can use the function *data* to load the example data or the basic R function *read.table* to load the custom data from local files. The function *GFFex* can be used to extract the information from a GFF3 format genome annotation file. Then, the function *ideogram* can be used to compute the information for all drawing elements based on the input files and generate a A4-sized SVG file containing a vector graphic which can be conveniently viewed and modified using the software Adobe Illustrator or Inkscape. Alternatively, users can also use the function *convertSVG* to convert this SVG file into an adjustable image format (pdf, png, tiff, or jpg) with a user-defined resolution according to the practical requirements.

In general, there are two types of data, i.e., continuous and discrete data. For mapping and visualizing, *RIdeogram* considers the continuous data, such as gene density across the whole genome in 1-Mb windows, as overlaid features and maps them on the idiograms with dark/light colors representing high/low values. For the other data type that are scattered throughout the whole genome, such as the chromosomal distribution of members in one gene family, *RIdeogram* can add track labels next to the idiograms with three shapes (box, circle and triangle) available to represent different characteristics of these members, such as the subclade that one gene member belongs to. Users can also combine the shapes and colors to represent more than three distinct characteristic types. Furthermore, users can also map the continuous data as a heatmap, a line or area chart along the idiograms. In addition, *RIdeogram* also provides functions for the visualization of dual and ternary genome synteny using Bezier curves on the idiograms.

*RIdeogram* is available through CRAN (https://cran.r-project.org/web/packages/RIdeogram/) and is developed on GitHub (https://github.com/TickingClock1992/RIdeogram). Further extensions in development and fixes can be seen in the issue listing page on the package's GitHub page. The new function that we are planning to implement in next version include, but are not limited to, developing more types of data visualization along the idiograms, visualizing genome synteny for more species and enlarging the

user-specified genome regions to display detailed characteristics, as we gather more from users.

## EXAMPLES

Our first example use the data contained in this package. After the completion of genome sequencing, assembly and annotation, *RIdeogram* can be used to give some idea of how genes are distributed across the whole genome. The example data contained numbers of protein-coding genes calculated in 1-Mb windows which can be considered as continues data and positions of 500 random selected non-coding RNAs, including ribosomal RNAs (rRNAs), transfer RNAs (tRNAs) and microRNAs (miRNAs), which can be considered as discrete data. *RIdeogram* maps the gene density information on the idiograms as overlaid features in a heat map and adds track labels next to the idiograms with green boxes, purple circles and orange triangles representing rRNAs, tRNAs and miRNAs, respectively (Fig. 1). Obviously, inter- and intra-chromosomal gene distributions are non-uniform. For instance, the chromosomal regions adjacent to the centromeres are gene-poor in chromosome 1, 9 and 16 while those are gene-rich in chromosome 11, 14 and 17. This function can be applied to many different situations, such as single nucleotide polymorphism (SNP) density and candidate markers (Fig. S1 & Data S1, original data see *Li et al., 2019*), DNA methylation dynamics and potential activated genes (Fig. S2 & Data S2, original data see *Huang et al., 2019*) and transcription factor (TF) binding sites and candidate target genes (Fig. S3 & Data S3, original data see *Shamimuzzaman & Vodkin, 2013*).

Besides visualizing some specific genome characteristics across the whole genome at the chromosome level as showed in Fig. 1, *RIdeogram* can also be used to compare two relevant genome features, such as gene and repeat density, which will provide some important implications for better understanding the relevance of chromosomal distribution patterns of these two features. The example data implemented in this package also contained the information of long terminal repeat (LTR) distribution across the human genome. Since the transposable elements have been suggested to have a potential detrimental effect on gene expression (*Hollister & Gaut, 2009*), the distributions of gene and LTR are supposed to be opposite across the whole genome as a result of natural selection. As expect, the region that has a relatively high gene content usually has a relatively low LTR density and vice versa (Fig. S4), indicating that LTR seems to avoid inserting in the regions with a high gene content in the genome. This similar phenomenon was also observed in the sunflower genome explained using two idiogram graphics, one showing the gene distribution and the other showing the LTR distribution (*Badouin et al., 2017*). Using *RIdeogram*, users can integrate these two graphics into one, much easier for researchers to interpret and readers to understand. Apart from the differences, this function can also be used to show the similarities, like the similar genetic diversity patterns across the whole genome between two geographical groups of the same species, in different label types (Fig. S5 & Data S4, Fig. S6, original data see *Chen et al., 2019*).

In addition, *RIdeogram* can also be used to show syntenic comparisons between two or three genomes. As shown in Fig. 2, the syntenic blocks between each pair of species,

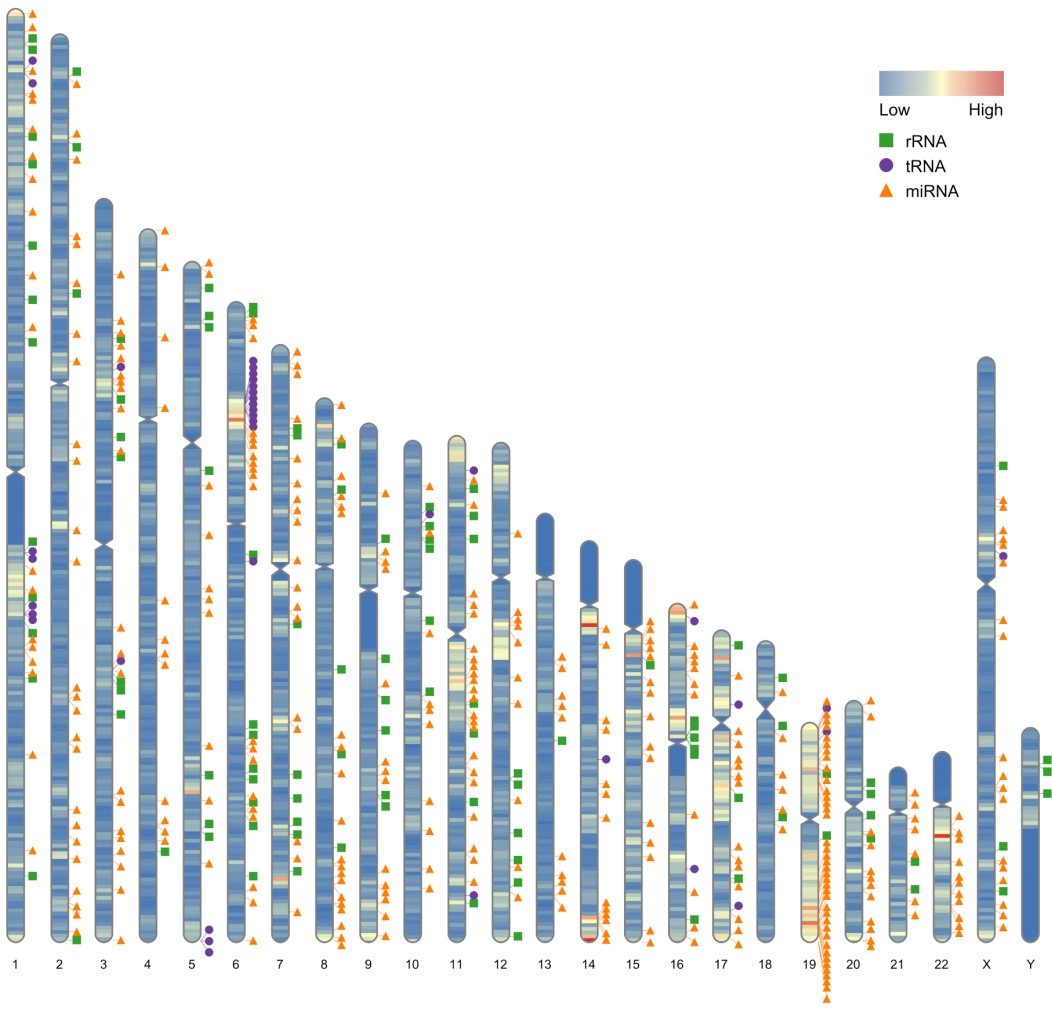

**Figure 1** **Gene distribution across the whole human genome.** The overlaid heatmap shows the gene density and the tack labels refer to 500 random selected RNAs consisted of rRNAs (green boxes), tRNA (purple circles) and miRNA (orange triangles) locus across the human genome. Annotation information was downloaded from the GENCODE website (https://www.gencodegenes.org).

which were identified using MCScan (*Tang et al., 2008*), were plotted. Particularly, a typical ancestral region in the basal angiosperm *Amborella* can be tracked to up to two regions in *Liriodendron* and to up to three regions in grape. Based on the fact that no lineage-specific polyploidy event has been found in *Amborella* and a whole-genome triplication has been detected in grape, it is reasonable to assume a single *Liriodendron* lineage-specific whole genome duplication event (*Chen et al., 2019*). Furthermore, *RIdeogram* allows to visualize a dual genome comparison, such as the genome synteny between human and mouse (Fig. S7 and Data S5). Compared to autosomes, the syntenic blocks between human and mouse X chromosomes occupy almost the entirety of each X chromosome, suggesting a highly conserved syntenic relationship of the X chromosome within the eutherian mammalian lineage (*Ross et al., 2005*).

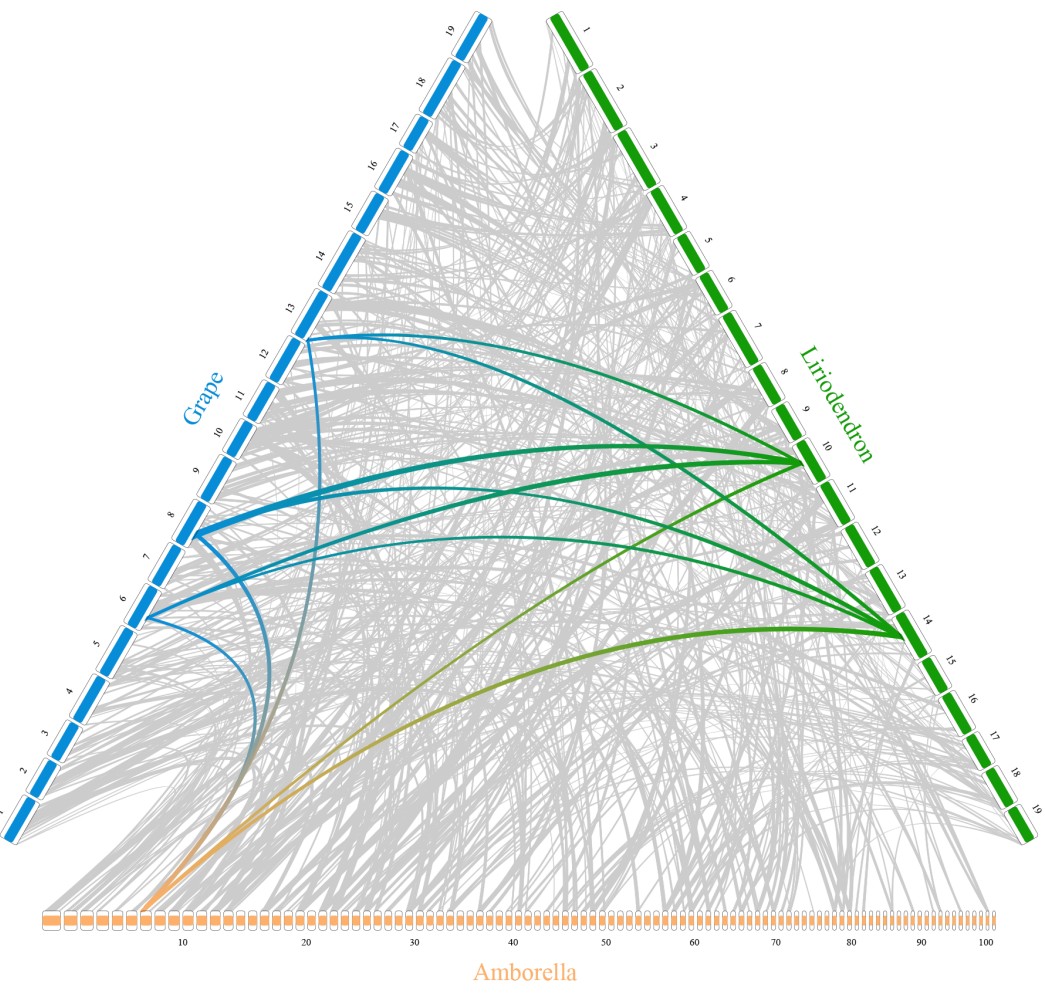

**Figure 2** **Syntenic comparison of three plant genomes.** Genome synteny patterns show that a typical ancestral region in the basal angiosperm *Amborella* can be tracked to up to two regions in *Liriodendron* and to up to three regions in grape. Gray wedges in the background highlight major syntenic blocks spanning more than 30 genes between the genomes (highlighted by one syntenic set shown in colored).

# CONCLUSION

The *RIdeogram* package provides an efficient and effective way to build idiograms with no species limitations and map genome-wide information on the idiograms for better visualizing and understanding the chromosomal distribution patterns of some particular genomic features. Meanwhile, this package can be also used to visualize syntenic analysis between genomes. Additionally, it is user-friendly and accessible for biologists without extensive computer programming expertise. Finally, *RIdeogram* can generate two types of images, a vector graphic or a bitmap file, both in high-quality and meeting conventional requirements for direct use in presentations or journal publications.

## ACKNOWLEDGEMENTS

We thank Dr. Zhongjuan Zhang for her comments on the manuscript.

### Funding

This work was supported by the Key Research and Development Plan of Jiangsu Province (BE2017376), the Foundation of Jiangsu Forestry Bureau (LYKJ[2017]42), the Qinglan Project of Jiangsu Province and the Priority Academic Program Development of Jiangsu Higher Education Institutions (PAPD). The funders had no role in study design, data collection and analysis, decision to publish, or preparation of the manuscript.

### Grant Disclosures

The following grant information was disclosed by the authors:
Key Research and Development Plan of Jiangsu Province: BE2017376.
Foundation of Jiangsu Forestry Bureau: LYKJ[2017]42.
Qinglan Project of Jiangsu Province.
Priority Academic Program Development of Jiangsu Higher Education Institutions (PAPD).

### Competing Interests

The authors declare there are no competing interests.

### Author Contributions

- Zhaodong Hao conceived and designed the experiments, performed the experiments, analyzed the data, performed the computation work, prepared figures and/or tables, authored or reviewed drafts of the paper, and approved the final draft.
- Dekang Lv performed the experiments, performed the computation work, authored or reviewed drafts of the paper, and approved the final draft.
- Ying Ge performed the experiments, performed the computation work, authored or reviewed drafts of the paper, typeset the code, and approved the final draft.
- Jisen Shi and Dolf Weijers performed the experiments, authored or reviewed drafts of the paper, and approved the final draft.
- Guangchuang Yu and Jinhui Chen conceived and designed the experiments, performed the computation work, authored or reviewed drafts of the paper, and approved the final draft.

### Data Availability

Data and codes are available at GitHub: https://github.com/TickingClock1992/RIdeogram.

## Supplemental Information

Supplemental information for this article can be found online at http://dx.doi.org/10.7717/peerj-cs.251#supplemental-information.

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
