# Peer review of "RIdeogram: drawing SVG graphics to visualize and map genome-wide data on the idiograms"

_PeerJ Computer Science, doi:10.7717/peerj-cs.251_

## Round 0.1 · original submission · Major Revisions

Try to express the features that differentiate your software of the state of the art. Also, it would be ideal to make a comparison with other software packages used for similar purposes.

Reviewer 1 ·

Basic reporting

Line 30-31: "As boundaries between model and non-model species are shifting" -> Model organisms are well defined, and have particular characteristics that make them be considered that way, not just their sequenced genome. The message is correct, but the manner is not appropriate.

Line 43-45: "many sequenced species have their genomes updated to chromosome level and more and more non-model species have their genomes sequenced" -> Redundant.

Line 49: "A ideogram" -> An idiogram.

Line 50: "a eukaryotic" -> an eukaryotic.

Idiogram and Ideogram is not the same, watch out!

There are more tools that offer the features described in this package and are not mentioned. Some examples are: KaryoPlotR, GenomeGraphs, RCircos, DRAWID, etc...

Need more References (look PDF)

To Include: I suggest including a comparison of this package with its main alternatives (look PDF)

If the strength of the package and the need it solves is to generate karyotypes of any species, it would be better to include one main-figure of a non-model species, not only human ideograms.

Experimental design

Line 56-58: "We still lack the choice of available drawing tools for plotting idiograms conveniently and effectively with a wide range species, despite the widespread need to visualize features along entire chromosomes" -> One of the packages mentioned in this article does the same and exactly the same way. Other packages such as KaryoPlotR also generate these types of figures and with many more features.

Validity of the findings

Despite specifying the need to generate ideograms due to the rise of new sequencing technologies, there is not direct connection between the results of these technologies and the package. It would be necessary to extract information from these results using programming tools or third parties. Although the flexibility provided by this tool makes it very useful and versatile, it is necessary to establish a connection between typical genomics files (Fasta, GTF/GFF, etc.) with the package.

At the current state of the package does not offer new options or greater utility than current alternatives. That is why I suggest the implementation of new functions for the package similars to present alternatives, providing the generation of SVG graphics and a different style as a differential element. For example, this package generates the same as ChromoMap, differentiating it only in the generation of the image and the style. But it is far from being a serious competitor of Idiogram.js or KaryoPlotR.

Annotated reviews are not available for download in order to protect the identity of reviewers who chose to remain anonymous.

Reviewer 2 ·

Basic reporting

Citation is missing for chromoMap (biorxiv), and I think karyoploteR, another non-overlapping but extensive package for plotting above ideograms, should be mentioned and added to the comparisons. Otherwise, the manuscript appears to meet all the standards of the journal.

Experimental design

Rideogram is different enough from other packages that draw ideograms, and is likely to be useful for exploratory and comparative genome-wide trait visualization.

Validity of the findings

The code is available on CRAN and github and the package includes clear documentation and reproducible examples.

Additional comments

This is a straightforward and user-friendly R package for drawing ideograms. I support the authors suggestion to add future extensions to the package and was satisfied to see responses to questions and issues raised on github. My one suggestion for an extension, while not at all necessary, is to add a function that aggregates raw data, by averaging or counting over a custom window-size, to produce the final “overlaid“ data. I believe this will increase usability among researchers without extensive computer programming expertise.

---

## Round 0.2 · Minor Revisions

Please follow the recommendations of the second reviewer. I think after this round the paper will be ready to be published

Reviewer 1 ·

Basic reporting

- Good use of the figures. They show the versatility of the package.

Experimental design

- It is easier to identify the strengths of the library by defining well the existing competitors. Good job.

Validity of the findings

- Good implementation of genomic files to the tool. I hope it is the first implementation of many more.

Reviewer 2 ·

Basic reporting

The 2nd introduction paragraph (line 51-68) is not clear. I think it would be more compelling if it was separated into these two paragraphs: 1. Feature comparison between Rideogram and other idiogram packages and 2. Novel use of Bezier curves in idiograms to express the relationship between genome-wide features and synteny.

The new option for plotting syntenic Bezier curves is a great addition and I expect many will find it useful. However, figure 2 does not include an idiogram although I expected it to show both, as an R package that plots idiograms with Bezier curves. I could also imagine that a dual-genome comparison figure of closely related species would be more useful as a comparative genomics example.

Syntenic Bezier curves are not a standard datatype. It would be helpful to add at least one citation of the software you used to generate the data (MCScan?), some documentation in the R package about the data structure, and maybe a wrapper function to handle the data.

The highlighted/colored Bezier curves in figure 2 extend beyond the range of the chromosome and would not align correctly with features in an idiogram. The plotting parameters should be fixed to accurately match the chromosomal coordinates.

Chromosome 19 in Fig S6 is missing a line graph.

Experimental design

The added feature of Syntenic Bezier curves in combination with idiograms is a novel feature and it would be worth expanding on why such a feature is needed/useful. For instance, there is some evidence that syntenic genes are more highly expressed and conserved. Rideogram could be a great tool to visualize this and also further explore potential relations between syntenic regions and other genomic features.

Validity of the findings

The implementation of line and bar charts makes Rideogram more comparable to existing tools. It remains to be a user-friendly and easily customizable R package for drawing high quality idiograms, which makes it a good alternative to existing packages. The addition of syntenic Bezier curves on top of the idiograms is a novel addition that could distinguish Rideagram from other idiogram-plotting packages.

Additional comments

Thank you for addressing the reviewer comments and for the additions you have made to the package. I think the addition of syntenic Bezier curves is a very useful addition that would fill a gap between idiogram and Circos plots. For this reason, I think it deserves to be described in more detail.

---

## Round 0.3 · accepted · Accept

The reviewers consider the paper is ready for publication now... and so do I.

Reviewer 2 ·

Basic reporting

Text and figures are clear, and the relevant background and citations are included.

Experimental design

The purpose of the package is well defined and does well in explaining the gap it aims to fill.

Validity of the findings

Rideogram has a comparable and useful set of features to existing tools, with the novel feature of syntenic bezier curves. It stands out as an easy to use and customize R package for genome-wide feature comparisons.

Additional comments

Thank you for addressing the comments. The article is in good shape and I am looking forward to future additions to Rideagram, especially those related to genomic-feature comparisons alongside synteny data.